# Foliar Mn and Zn Treatments Improve Apple Tree Nutrition and Help to Maintain Favorable Soil pH

Andrei I. Kuzin [1,2,3,](), Natalia Ya. Kashirskaya [1], Alexei E. Solovchenko [3,4](), Alexei V. Kushner [1], Anna M. Kochkina [1], Ludmila V. Stepantzova [2] and Vyacheslav N. Krasin [2]

1   I.V. Michurin Federal Scientific Centre, 393774 Michurinsk, Russia; kashirskaya@fnc-mich.ru (N.Y.K.); alexkoushner@mail.ru (A.V.K.); ms.anna.step@mail.ru (A.M.K.)
2   I.V. Michurin Institute of Fundamental and Applied Agrobiotechnologies, Michurinsk State Agrarian University, 393760 Michurinsk, Russia; stepanzowa@mail.ru (L.V.S.); krasin84@yandex.ru (V.N.K.)
3   Institute of Natural Sciences, Derzhavin Tambov State Agrarian University, 393760 Tambov, Russia; solovchenkoae@my.msu.ru
4   Faculty of Biology, Lomonosov Moscow State University, 119234 Moscow, Russia
*   Correspondence: andrey.kuzin1967@yandex.ru

**Abstract:** The foliar application of micronutrients can improve primary nutrient uptake. As a result, foliar treatments can reduce fertilizer application rates and help to maintain the natural health of soil. Here, we report on the tentative implementation of this approach in an apple orchard located in a temperate climate (JSC "Dubovoye" 52°36′57.1″ N 40°17′04.1″ E; planted in 2002 according to the 6 × 4 m or 417 trees ha$^{-1}$ cultivar (cv.) Bogatyr grafted on B118 (Budagovskii 118). Manganese treatments augmented foliar nitrogen content and, in certain seasons, foliar phosphorus, whereas zinc treatments enhanced foliar potassium. Low-rate chemical fertilizers application (once in 5 years) on the background of initial high-rate organic fertilization (60 t ha$^{-1}$ manure) allowed us to retain the optimal soil pH in the experimental orchard.

**Keywords:** apple tree; foliar nutrients; fruit yield; micronutrients; nitrogen; phosphorus; potassium; soil acidity





## 1. Introduction

High-density orchard planting has remained the mainstream approach to apple fruit production for several decades, mainly for land and labor shortages driven by the need to increase the areal productivity and profitability of orchards [1–3]. These goals are achieved through drastic intensification of the production process including an increase in planting density, which elevates the yield but also expenses (planting material, tree support, fertigation, etc.) [4,5]. Although the profitability of such orchards is high, their estimated lifespan does not exceed 20 years [6].

In particular, the need for irrigation stems from the fact that the tree roots do not penetrate deep into the soil in such orchards. Svoboda et al. [7] found that the primary roots occupy the top 20 cm of soil when it is irrigated, but some roots can reach as deep as 80 cm. The bioavailable nutrients in the soil are either taken up by the roots or washed out of reach of the roots. For instance, calcium, magnesium, potassium, sulfur, and boron decreased in the orchard soil layer of 0–15 cm after three years of irrigation; pH and electrical conductivity (eC) values also declined [8]. Paradoxically, fertigation can lead to nutrient loss and their leaching to the groundwater [9–11]. Acidic soil pH decreases nutrient availability, requiring higher application rates for fertilizers; this causes further groundwater pollution and so on [8,10,12]. This makes high-density orchards hardly sustainable and also unfriendly to the environment [13].

In the last 35 years, the intensification of agriculture has increased the application rate of fertilizers by almost 7 times for nitrogen and 3.5 times for phosphorus [14]. The trend of

boosting apple productivity by increasing the application of chemicals including fertilizers in high-density orchards has to be changed to the opposite trend of a reduction in the application of chemicals to save the environment [15]. This increases the environmental safety and sustainability of orchards but at the cost of a reduction in yield and hence the loss of the most important economic advantage of high-density orchards—their high profitability.

Because of land and labor shortages, apple fruits are mostly grown in high-density orchards with drip irrigation and fertigation. This practice impacts soil health, especially its pH. On the other hand, conventional apple orchards with expansive canopies remain promising. Thus, orchards with a planting density of 800–1000 trees ha$^{-1}$ (tr. Ha$^{-1}$) are more resilient to the climate fluctuations of Central Russia. They do not need drip fertigation since their roots span a larger soil volume than those of trees in high-density orchards. Another approach to enhance the efficiency of fertilization is lifting possible limitations for macronutrient uptake by enhancing micronutrient supply to the trees.

In view of what is said above, it would be interesting to study the nutritional patterns underlying the sustainability of "conventional" orchards with expansive canopies whose roots span the bulk of fertile soil in these orchards. For instance, the roots of apple trees grafted on B118 rootstock reach down to 1–1.5 m deep and 3–4 m around (if the planting pattern permits), and the main part of the root system is in the soil layer that is 0.2–0.8 m deep and 1.0–1.5 m around (Figure S1) [16,17]. Therefore, trees grafted on B118 rootstocks grow well in soil with relatively low fertility and possess resistance to fire blight [17].

Chernozem soil is rich in macronutrients; these soils also feature an efficient microbiome, increasing the availability of nutrients to plants [18–20]. These soil features could be used for the improvement of apple tree nutrition. Soil frequently contains a lot of phosphorus, potassium, and calcium since these are elements present in the parent rock of the soil [21,22]. Thus, the availability of nutrients for plants can be low. It is determined by several factors, including soil acidity, cation exchange capacity, base saturation, and soil aggregates [23]. Frequent fertigation can impair these properties of the soil and reduce nutrient availability [8,10,24,25]. There are many reports in the literature suggesting that reducing fertilizer application rates helps to retain soil health [26–28].

Foliar fertilization is a fertilization technique based on the application of nutrient solution by spraying it onto plant leaves. One of the most significant advantages of foliar fertilization is the application of nutrients close to the site of their uptake and diminishing negative effects on the soil like acidification, etc. [29,30]. Although it is not possible to supply all of the needed nutrients through the leaves, conventional soil fertilization is still needed. A decrease in rates of fertilizer application to the soil and an increase in foliar fertilization reduce the wasting of nitrate N and P in the 0–20 cm soil layer [31].

There are also reports on the stimulatory effects of foliar fertilization on the uptake of soil nutrients. Thus, the application of ammonium N enhanced uptake from soil $^{15}$N by cotton plants [32]. Foliar application of Fe compounds on pepper stimulated an increase in P content in the shoots and roots and K, Mg, and Ca contents in the shoots [33]. The application of Cu-based foliar fertilizer with added Zn and controlled-release urea promoted plant growth and soil mineral N absorption [34]. The combination of soil and leaf fertilization is a promising method to reduce the utilization of N, P, and K. Foliar $ZnSO_4$ combined with macronutrient fertilization can reduce the conventional N application rate by 15% [35]. Foliar fertilizer application after soil fertilization is an effective method to increase the contents of trace elements in crops and crop yield and to improve the soil environment [36]. Therefore, foliar fertilization is an effective measure to improve the soil environment and crop quality, especially under restricted soil nutrient utilization and high soil nutrient loss rates [37].

Interesting results obtained in recent studies suggest that foliar fertilization can improve the mineral status of plants by decreasing soil fertilizer application rates [38–40]. In the present study, we tested the approach by which micronutrient foliar fertilization increases macronutrient uptake. We hypothesized that in this case (i) no macronutrient

fertilization will be required and (ii) the macronutrient demand will be covered, at an optimal pH value, by the broadly spread root system.

## 2. Materials and Methods

### 2.1. Location and Soil Properties

The study, which was conducted in JSC "Dubovoye ($52°36'57.1''$ N $40°17'04.1''$ E), spans three vegetation seasons (2020, 2021, and 2022). The orchard was planted in 2002; upon planting, composted cow manure was applied at a rate of 60 t ha$^{-1}$, and the soil was plowed (45 cm deep). The orchard planting pattern was 6 × 4 m (417 tr. ha$^{-1}$). The drip irrigation system was installed on the plantation. The soil between the rows was sown with a bean–cereal mixture. The drip tapes were placed on the ground. Chemical fertilizers were applied, at a low rate, once every 4–5 years (last time in spring 2021). Complete fertilizer "ammophoska" (($NH_4)_2SO_4$ + ($NH_4)_2HPO_4$ + $K_2SO_4$) was used; the application rate was $N_{12}P_{15}K_{15}S_{14}Mg_{0.5}Ca_{0.5}$. The soil is typical chernozem on carbonate loam. The soil supply with primary nutrients can be characterized as high (Table 1).

**Table 1.** Primary nutrient contents (at the beginning of the study).

| Soil Layers (cm) | N Hydrolysable (mg kg$^{-1}$) | P Available (mg kg$^{-1}$) | K Exchangeable (mg kg$^{-1}$) | Ca Exchangeable, (mg kg$^{-1}$) | Humus |
|---|---|---|---|---|---|
| 0–20 | 145.3 | 173.0 | 132.9 | 3883 | 5.7 |
| 21–40 | 135.4 | 156.9 | 117.6 | 3787 | 5.4 |
| 41–60 | 106.0 | 99.6 | 96.7 | 3803 | 4.5 |
| 61–80 | 79.8 | 82.2 | 102.9 | 3815 | 2.4 |

The field moisture capacity of the meter layer is 28.6–29.9% and the sum of the absorbed bases (Cation Exchange Capacity) is 29.1–31.4 meg/100 g of the soil of the 0–40 cm layer.

Foliar fertilization was carried out with a sprayer SLV-2000 (ITA "MECAGRO", Chisinau, Moldova), with 400 L of the working solution per treatment. The experimental rows were interleaved with two shielding rows.

The soil profile from the experimental orchard is presented in Figure 1.

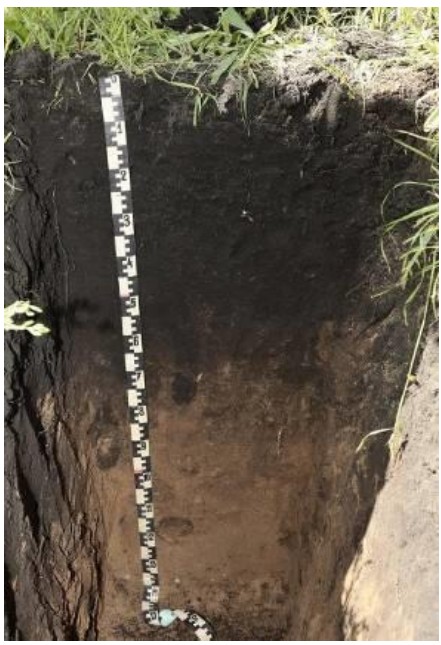

**Figure 1.** Soil profile in the experimental orchard.

Soil—heavy loam leached medium-sized meadow-chernozem on the cover carbonate loam.

Ap (0–20 cm)—moist, dark gray, medium compacted, fine-grained, heavy loam, coprolites and plant roots are present, the border is wavy, and the transition is clear.

App (20–37 cm)—moist, dark gray, medium compacted, lumpy-prismatic, heavy loam, coprolites and plant roots are present, the border is wavy, and the transition is clear.

A1 (37–67 cm)—moist, dark gray, brownish at the bottom, medium compacted, fine-grained, heavy loam, numerous coprolites and plant roots are present, the border is lingual, and the transition is clear.

AB (67–82 cm)—moist, mottled color with humus stains, fine-grained, heavy loam, coprolites and plant roots are present, the border is lingual, and the transition is clear.

B1g$^{/}$ (82–120 cm)—moist, light brown, dense, lumpy-granular, heavy loam, coprolites are found, weakly glued humus cutans are found, plant roots are present, the border is smooth, the transition is clear.

B2Ca (120–135 cm)—moist, light brown, dense, lumpy-prismatic, weakly structured, heavy loam, boils from HC1, carbonates are present, the wound is smooth, and the transition is gradual.

CCa (135–150 cm)—moist, light brown, dense, lumpy-prismatic, weakly textured, heavy loam, boils violently from HC1, abundant carbonate mycelium, carbonate deposits, carbonate concretions 1–3 cm (very dense, with internal recrystallization carbonate nodules), and rare inclusions of peeled fine-grained (spots of bluish color).

### 2.2. Experimental Design

The experimental plot comprised 63 trees per row. Each treatment (4 replicas; 15 trees each) was placed on a separate row (Table 2).

**Table 2.** Experimental design.

| Treatments | Symbol |
| --- | --- |
| 1. Control (no foliar fertilization) | C |
| 2. Ca | T1 |
| 3. Zn, Cu | T2 |
| 4. Fe, Mn | T3 |
| 5. Full program of foliar fertilization (Micronutrients + amino acids + calcium) | T4 |

We applied for foliar sprayings agrochemicals manufactured by "Schelkovo Agrohim" (Scheljovo, Russia) (Table 3).

**Table 3.** Foliar spraying program.

| Growth Stage | Agrochemicals and Application Rate | Date of Spraying |
| --- | --- | --- |
| | T1 | |
| 56 (Green bud stage: single flowers separating (still closed)) | Ultramag calcium (UlCa) (Ca 17%, N 10%, MgO 0.5%) 2.0 L ha$^{-1}$ | 04.29.2020; 05.05.2021; 05.02.2022 |
| 69 (End of flowering: all petals fallen) | UlCa 2.0 L ha$^{-1}$ | 05.22.2020; 05.23.2021; 05.29.2022 |
| 72 (Fruit diameter up to 20 mm) | UlCa 3.0 L ha$^{-1}$ | 06.06.2020; 06.01.2021; 06.13.2023 |
| 74 (Fruit diameter up to 40 mm) | UlCa 3.0 L ha$^{-1}$ | 06.12.2020; 06.14.2021; 06.24.2022 |
| 75 (Fruit above half final size) | UlCa 3.0 L ha$^{-1}$ | 06.27.2020; 06.28.2021; 07.04.2022 |
| 76 (Fruit about 60% final size) | UlCa 3.0 L ha$^{-1}$ | 07.06.2020; 07.15.2021; 07.16.2022 |
| 77 (Fruit about 70% final size) | UlCa 3.0 L ha$^{-1}$ | 07.27.2020; 07.25.2021; 08.05.2022 |
| 78 (Fruit about 80% final size) | UlCa 3.0 L ha$^{-1}$ | 08.14.2020; 08.11.2021; 08.15.2022 |
| | T2 | |
| 56 (Green bud stage: single flowers separating (still closed)) | Ultramag Chelate Zn-15—(ChZn) 1.0 L ha$^{-1}$ | 04.29.2020; 05.05.2021; 05.02.2022 |
| 74 (Fruit diameter up to 40 mm) | ChZn 1.0 L ha$^{-1}$ | 06.12.2020; 06.14.2021; 06.24.2022 |
| 75 (Fruit above half final size) | Ultramag chelate Cu-15 (ChCu) 0.5 L ha$^{-1}$ | 06.27.2020; 06.28.2021; 07.04.2022 |
| 77 (Fruit about 70% final size) | ChCu 0.5 L ha$^{-1}$ | 07.27.2020; 07.25.2021; 08.05.2022 |

**Table 3.** *Cont.*

| Growth Stage | Agrochemicals and Application Rate | Date of Spraying |
|---|---|---|
| | T3 | |
| 56 (Green bud stage: single flowers separating (still closed)) | Ultramag chelate Mn-13 (ChMn) (Mn 13%) 0.5 L ha$^{-1}$ | 04.29.2020; 05.05.2021; 05.02.2022 |
| 72 (Fruit size up to 20 mm) | ChMn 1.0 L ha$^{-1}$ | 06.27.2020; 06.28.2021; 07.04.2022 |
| 74 (Fruit diameter up to 40 mm) | Ultramag chelate Fe-13 (ChFe) 1.0 L ha$^{-1}$ | 06.12.2020; 06.14.2021; 06.24.2022 |
| 77 (Fruit about 70% final size) | ChFe 1.0 L ha$^{-1}$ | 07.27.2020; 07.25.2021; 08.05.2022 |
| | T4 | |
| 56 (Green bud stage: single flowers separating (still closed)) | Biostim Growth (BG) (aminoacids 4%, N 4%, P$_2$O$_5$ 10%, MgO 2.0%, SO$_4$ 1.0%, Fe 0.4%, Mn 0.2%, Zn 0.2%, B 1%) 1.0 L ha$^{-1}$; Ultramag Boron (Bor) (B 11%, N 3.5%) 1.0 L ha$^{-1}$; UlCa 2.0 L ha$^{-1}$, | 04.29.2020; 05.05.2021; 05.02.2022 |
| 61 (Beginning of flowering: about 10% of flowers open) | BG 1.5 L ha$^{-1}$, Bor 1.0 L ha$^{-1}$ | 05.07.2022; 05.14.2021; 05.16.2022 |
| 57 (End of flowering: all petals fallen) | UlCa 2.0 L ha$^{-1}$; BG 2.0 L ha$^{-1}$ | 05.04.2020; 05.22.2021; 05.29.2022 |
| 72 (Fruit size up to 20 mm) | UlCa 3.0 L ha$^{-1}$ Ultramag Potassium (ULK) (K$_2$O 22%, N 3.5%) 3.0 L ha$^{-1}$ | 06.06.2020; 06.01.2021; 06.13.2023 |
| 74 (Fruit diameter up to 40 mm) | ULK 3.0 L ha$^{-1}$; UlCa 3.0 L ha$^{-1}$, BG 1.0 L ha$^{-1}$ | 06.12.2020; 06.14.2021; 06.24.2022 |
| 75 (Fruit above half of the final size) | UlCa 3.0 L ha$^{-1}$; Biostim Development (BD) (N 2.0%, CaO 10.0%, MgO 5.0%, Fe 0.7%, Mn 2.0%) 2.0 L ha$^{-1}$ | 06.27.2020; 06.28.2021; 07.04.2022 |
| 76 (Fruit about 60% of the final size) | UlCa 3.0 L ha$^{-1}$ | 06.27.2020; 06.28.2021; 07.04.2022 |
| 77 (Fruit about 70% of the final size) | UlCa 3.0 L ha$^{-1}$ | 07.27.2020; 07.25.2021; 08.05.2022 |
| 78 (Fruit about 80% of the final size) | UlCa 3.0 L ha$^{-1}$ | 08.14.2020; 08.11.2021; 08.15.2022 |

The concentrations of the working solutions for foliar nutrients were determined according to the manufacturer's (Schelkovo Agrohim, Russia) recommendations. We applied separate micronutrients in our trial twice a season according to the foliar fertilization guide [41]. The growth stages were indicated according to the BBCH Monograph [42].

The treatments were scheduled based on the plant protection scheme and the predicted presence of scab and codling moths (for details, see Supplementary Materials).

*2.3. Sampling and Assays*

Leaf samples were taken from the middle of annual shoots from the mid-height of the canopies. We sampled four leaves per tree in the middle of the canopy from the middle of annual shoots from eight random trees from each side. Subsequently, we selected 32 leaf sub-samples for each replica, constituting a pooled sample that was used for the analyses. The leaves were also sampled 1 h before spraying to provide a baseline for the assessment of the foliar fertilization effect. Soil samples were taken at a 0.3–0.5 m distance from tree trunks at the same points throughout the entire research period. We determined the following nutrients in the leaves: nitrogen (N) (Kjeldahl method, AKV-20, JSC Villlitek, Moscow, Russia); phosphorus (P) (photometric molybdenum blue method, Hitachi U2000, Hitachi, Ltd., Tokyo, Japan); iron (Fe) (photometric method with sulfosalicylic acid and ammonia, Hitachi, Ltd., Hitachi U2000, Tokyo, Japan); potassium (K), calcium (Ca) (flame photometric method, FPA-2.01, JSC ZOMZ, Zagorsk, Russia), copper (Cu), zinc (Zn), manganese (Mn), and molybdenum (Mo) (by atomic absorption spectroscopy, MGA-915MD, Lumex, Saint-Petersburg, Russia). The soil analyses were carried out in the agrochemical laboratory of I.V. Michurin Federal Scientific Centre. The soil nitrogen was assayed by the Kjeldahl method, phosphorus was assayed in the extracts made with 0.5N CH$_3$COOH followed by the phosphomolybdenum blue method and SnCl$_2$ staining and detection at 750 nm, total exchangeable potassium was assayed by a flame photometer, exchangeable soil calcium was determined by the complexometric method (titration of calcium with Trilon B at pH 12.5–13.0 using murexide as an indicator), and pH$_{KCl}$ was determined on an Expert-001 pH meter, Econics-Expert, LTD., Moscow, Russia [43]. Since the maintenance of soil was the same for all of the experimental treatments, we did not take individual soil samples from

the plots of each treatment but sampled the soil in five locations and averaged the results. In 2020, soil sampling was carried out only once (for the determination of soil acidity on 14 August 2020). In 2021 and 2022, soil samples were taken five times per season.

### 2.4. Statistic Treatment

The data were analyzed according to Fisher's method [44]. We calculated the least significant difference (LSD) between the various treatments at $p < 0.05$. The differences that were higher than the computed LSD value were considered to be significant. The period that we took for correlation evaluation started from the first investigated micronutrient treatment and included 2–3 weeks after the last. We also calculated the coefficient of determination.

## 3. Results

### 3.1. Foliar Macronutrient Content

#### 3.1.1. Nitrogen (N)

There was no foliar application of macronutrients including N, in particular, to avoid the inhibition of N uptake via the roots (see [45,46]). In Figure 1, the results of the two treatments are presented, which showed the largest effect (the results for other treatments can be found in the Supplementary Materials).

Foliar N in all treatments was relatively high in 2020 (Figure 2a). A significant increase in foliar N was observed in T3 (56; 72 (growth stages according to the BBCH [42]—Mn treatments, 74; 77—Fe treatments) and T4 (the variant with the complete foliar fertilization program). Although in the case of good overwintering, high foliar N can, in principle, result from the mobilization of N reserves stored in other tissues, the timing of foliar spraying suggests that such an effect could be caused by Mn spraying. Later, the foliar N decreased (especially in T4), but this was a general trend for all experimental variants.

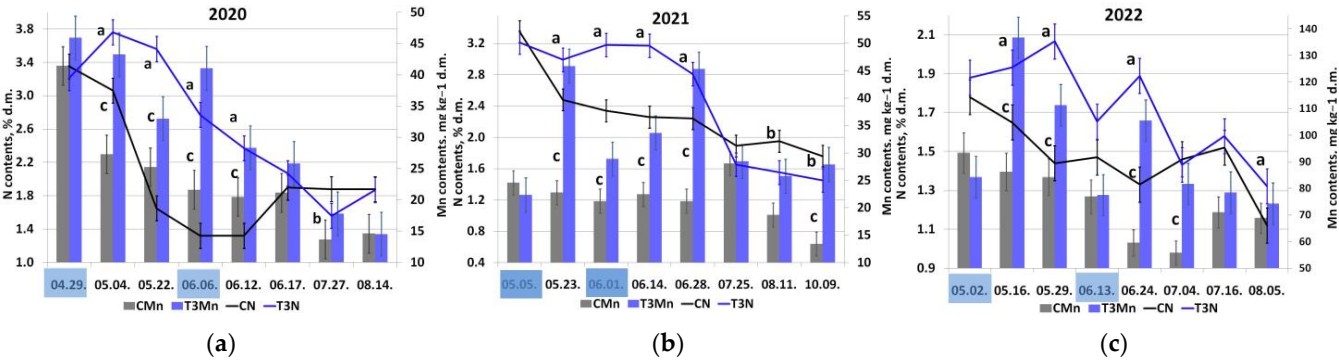

**Figure 2.** Seasonal changes in foliar N (lines, left scale) under the influence of manganese treatments (bars: foliar Mn, right scale) in the seasons of (**a**) 2020, (**b**) 2021, and (**c**) 2022. The dates of spraying are highlighted in blue (see also Figures S1–S3 and Tables S1–S15). d.m.—dry matter. The letters a and b—significant differences between lines, and the letter c—significant differences between bars.

In 2021, foliar N was high (Figure 2b) and also declined towards the end of the season. Mn spraying (T3) stimulated foliar N increase from the end of May until the end of June. Starting in June, Fe preparations were included in the T3 treatment. Taking into account the timing of foliar fertilization, it is more likely that the overall effect of this treatment was caused by the Mn included in the spraying from the very beginning. In 2022, foliar N peaked at the end of May and in the middle of June 2022. Likely, the observed effects reflected the overlapping of the foliar Mn spraying with the period when the N demand of the tree was the highest (Figure 2c).

There was a strong correlation between foliar N and Mn contents in certain treatments (Figure 3). Interestingly, in the control, these relationships were even stronger than in the experimental treatments. The peak values of N and Mn departed from the common

trend of the corresponding treatment, likely due to the lag between the peaks of Mn after the spraying and the following increase in foliar N. The strongest foliar Mn vs. foliar N correlation was observed in May–June. This indicates that the most pronounced effect of foliar fertilization can be expected in the periods of the highest micronutrient demand by the plant.

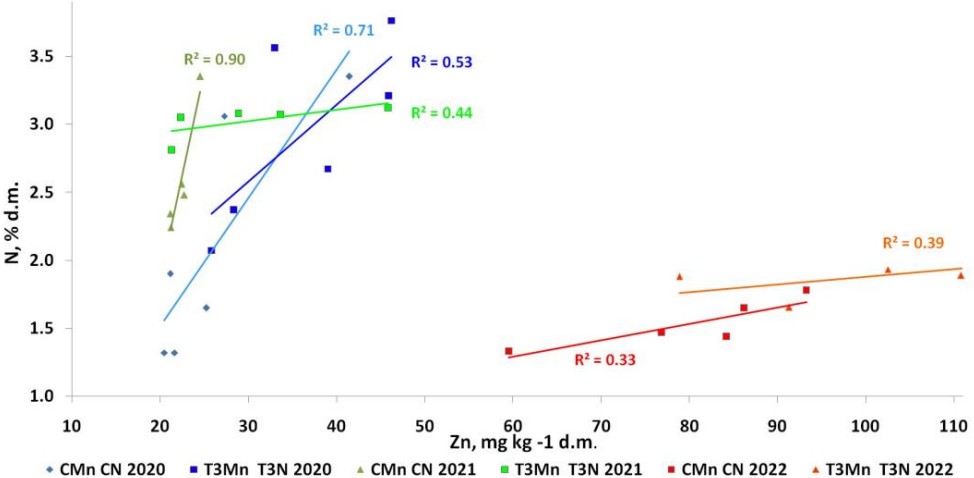

**Figure 3.** The relationships between foliar Mn and N recorded in the different treatments (indicated on the plot; see also the Methods section).

### 3.1.2. Phosphorus (P)

Foliar P was relatively high in May 2020 (Figure 4a), with it being the highest in the T4 treatment (see also Figure S4). Later, foliar P decreased in all experimental variants, with it remaining in the range of 0.2–0.4% d.m. with some fluctuations. These circumstances complicated the pinpointing of foliar spraying, which boosted the uptake of P by the plant roots in 2020.

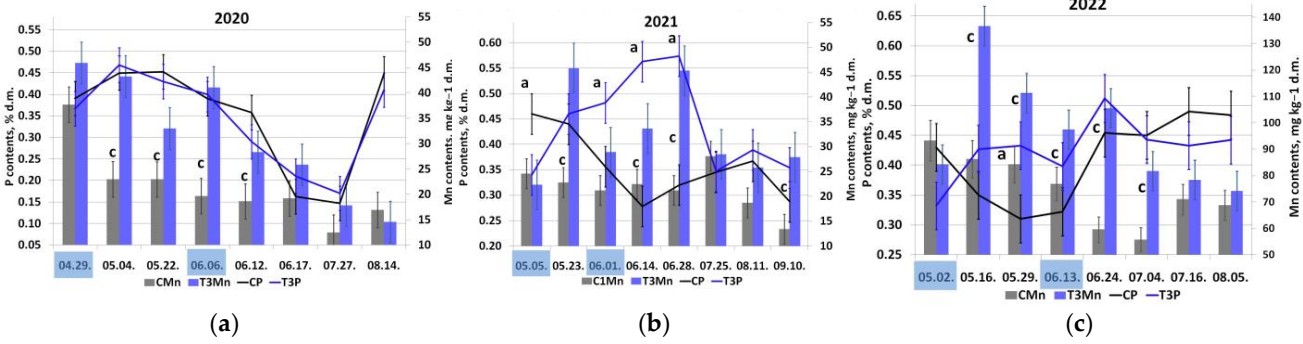

**Figure 4.** Seasonal changes in foliar P (lines, left scale) under the influence of manganese treatments (bars: foliar Mn, right scale) in the seasons (**a**) 2020, (**b**) 2021, and (**c**) 2022. The dates of spraying are highlighted in blue (see also Figures S4–S6 and Tables S1–S15). d.m.—dry matter. The letter a—significant differences between lines, and the letter c—significant differences between bars.

Foliar P varied significantly in all the experimental variants in 2021 (Figure 4b and Figure S5) but increased considerably in the T3 variant from the beginning of May until the end of June. A similar increase in foliar P was recorded in T4 during this period, but its magnitude was significantly lower than that in T3. Importantly, the increase in foliar P was observed only in the treatments with Mn foliar spraying. In 2022, the largest increase in foliar P phosphorus occurring in May was recorded in T4 (Figure S6); it even surpassed the upper optimum limit for apples [47]. We also observed an increase in foliar P in T3 (Figure 4c). As a result, the optimal P content was reached during bloom, which is very

important for the fruit set. Overall, Mn spraying is likely a plausible measure for increasing P uptake.

The foliar P contents had a strong upward trend, while the foliar Mn increased in the control ($R^2_{CMnCP2022}$ = 0.38) but especially after Mn treatments ($R^2_{T3MnT3P2021}$ = 0.88) in 2021 (Figure 5). The correlation was not so strong in 2022. In this year, the increase in leaf phosphorus contents was recorded only after Mn spraying. ($R^2_{T3MnT3P2022}$ = 0.21).

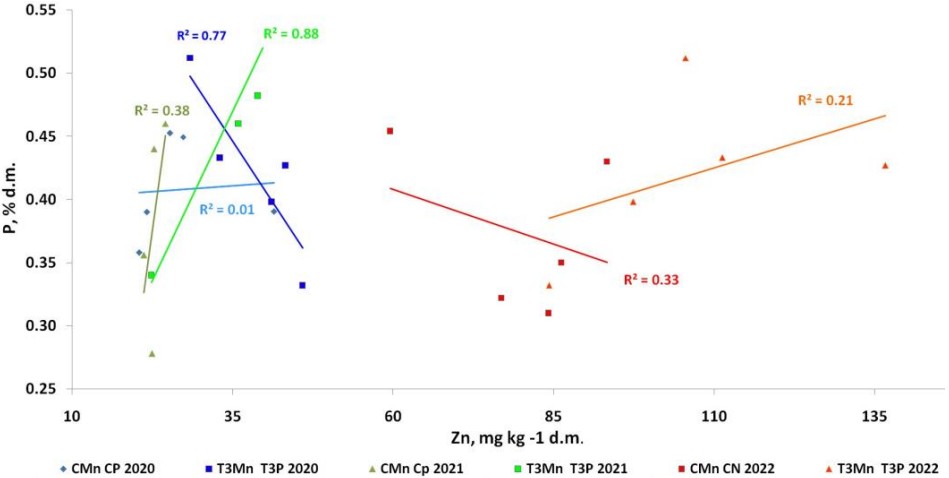

**Figure 5.** The relationships between foliar Mn and P recorded in the different treatments (indicated on the plot; see also the Methods section).

### 3.1.3. Potassium (K)

In 2020, changes in foliar K showed neither a discernible pattern (Figure 6a) nor correspondence to the data published in the literature [48]. None of the treatments was significantly different from each other in May. Foliar K decreased sharply to the middle of June, returning in several experimental variants (C (Control) and T4 (Figure S7) to the level recorded at the beginning of May. Foliar K in the T1 (the complete calcium foliar sprays), T2 (T2—56; 74 foliar Zn spray; 75; 77 foliar Cu spray), and T3 variants was also low but still significantly higher than in other variants. From June to the beginning of July, foliar K peaked in the C and T4 variants and declined in the middle of July, with it peaking again at the end of July. Overall, the effect of foliar treatments on the leaf K level was vague, although a relatively stable increase in foliar K was noted in the variants with Zn spraying (T2 and T4).

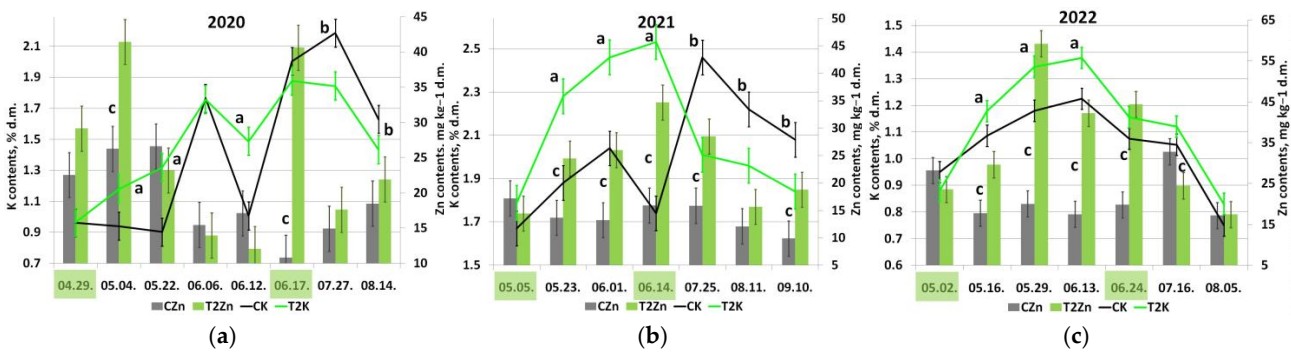

**Figure 6.** Seasonal changes in leaf potassium (lines, left scale) under the influence of manganese treatments (bars: foliar Zn, right scale) in the seasons (**a**) 2020, (**b**) 2021, and (**c**) 2022. The dates of spraying are highlighted in green (see also Figures S7–S9 and Tables S1–S15). d.m.—dry matter. The letters a and b—significant differences between lines and the letter c—significant differences between bars.

In the season 2021, the pattern of foliar K changes also did not correspond to the one documented in the literature [48–50]. Until mid-August, the foliar nutrient content

was high in the variants C (Figure 6b) and T4 (Figure S8) but started to decline thereafter. Likewise, the foliar K increased until the middle of July and decreased thereafter in the variant T1. The foliar K content in the variant T2 increased significantly from the beginning of May until mid-June, likely driven by the Zn-containing spraying.

In 2022, foliar K increased in all experimental treatments from the beginning of the season to mid-June and then decreased (Figure 6c and Figure S9). There were no significant differences between the treatments with a single exception of treatment T2 which displayed the distinct maximum of foliar K on 13.06.22 (likely due to the foliar Zn spraying as well).

The information about the influence of microelements on foliar K is scarce; its studies are complicated by the different mobility of mineral nutrients in plants. The most spectacular results in our study were obtained for Zn spraying. Though in 2020 we did not see a relationship between Zn treatments and foliar K, this relationship was evident in 2021 and 2022 (Figure 7). In 2021 and 2022, the foliar Zn spraying boosted the foliar K; this relationship was not observed in the control treatment in all seasons of the study.

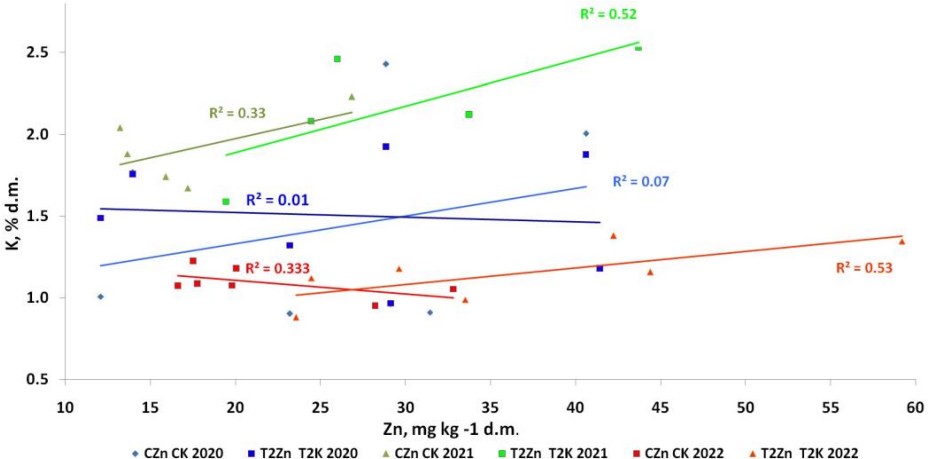

**Figure 7.** The relationships between foliar Zn and K recorded in the different treatments (indicated on the plot; see also the Methods section).

### 3.2. Soil pH

The average $pH_{KCl}$ value was 6.70; it varied within the range of 6.49–6.93 in 2020. We studied seasonal pH changes after the application of the ammophoska fertilizer (which was applied directly after snowmelt in April 2021; Figure 8).

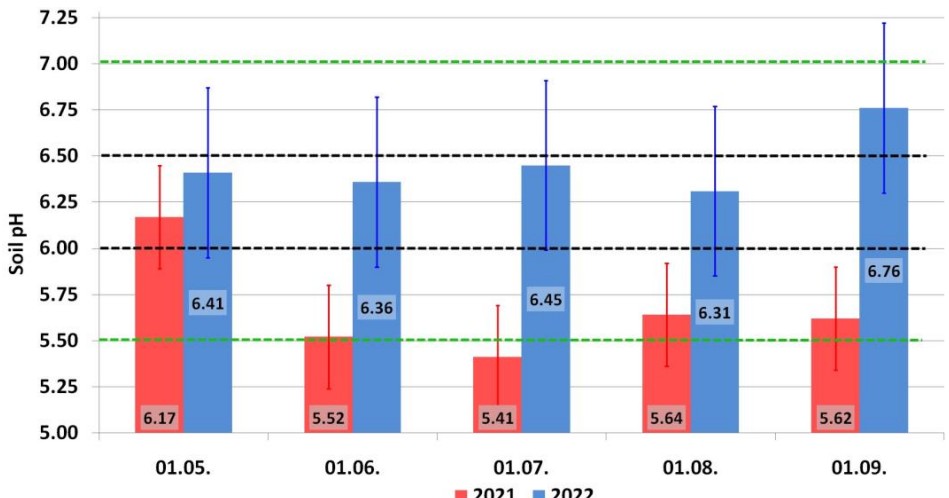

**Figure 8.** The effect of chemical fertilizer application in 2020 and 2021 on the seasonal changes in soil pH. Black dotted lines—the optimal pH range for N and K uptake; the green dotted lines—the optimal pH range for P uptake.

Soil acidity increased (the soil pH declined) on 1 May 2021 as compared to 14 August 2020. Later in 2021, soil acidity increased until 1 July 2021, when the lowest pH value was recorded. The pH values declined slightly until 1 August 2021 and remained at this level until 1 September 2021. The soil was mildly acidic, so the pH was favorable for the uptake of macronutrients in 2022. Attention must be paid to further increases in $pH_{KCl}$, which took place by the end of the season. The $pH_{KCl}$ variation after chemical fertilization was relatively high (pH 5.53–7.10), impacting the significance of the difference in the values documented in 2022.

### 3.3. Yield

The lowest bulk yield was documented in the control (C) lacking the foliar spraying (Table 4). In 2020, the largest yield was in the variant T1. May was cool in that season with a period of waterlogging affecting the outcome of the Ca treatments (Supplementary Materials), especially those carried out during the bloom for improving the fruit set [51]. Ca plays a vital role in plant growth and development as well as in stress responses as an intracellular messenger [52,53]. Obviously, Ca foliar spraying improved the plant's resistance to stress during periods of unfavorable weather. The variants T3 and T4 revealed no significant differences to T1 despite the lower yield.

**Table 4.** The effect of the treatments on yield, kg $tr^{-1}$.

| Treatments | 2020 | 2021 | 2022 | AVG | Total |
|---|---|---|---|---|---|
| C | 38.1 [a] | 33.9 [a] | 39.8 [a] | 37.3 [a] | 111.8 [a] |
| T1 | 44.8 [b] | 32.1 [a] | 46.9 [b] | 41.3 [b] | 123.8 [b] |
| T2 | 39.8 [a] | 31.3 [a] | 51.6 [c] | 40.9 [b] | 122.7 [b] |
| T3 | 41.6 [b] | 32.7 [a] | 52.6 [c] | 42.3 [b] | 126.9 [b] |
| T4 | 42.1 [b] | 33.4 [a] | 55.0 [c] | 43.5 [b] | 130.5 [b] |
| $LSD_{05}$ | 3.4 | 2.3 | 4.2 | 3.3 | 8.6 |

Different letters indicate a significant difference between the treatments.

The yield was lower in all treatments in the season 2021, likely due to the summer drought (Supplementary Materials). Drought is one of the major abiotic stresses reducing fruit yields [54]. Cv. Bogatyr was bred in 1926 but began to be planted in commercial orchards in 1972 [55]. This cultivar possesses relatively high frost hardiness but it is vulnerable to drought. The highest yield was in the control (C), and in the treatment T4 it was nearly the same. There were no significant differences between all the treatments in this year regarding the fruit yield. The application of ammophoska to the soil exerted no measurable effect on the yield in the background of harsh weather. Kowalczyk et al. [56] showed that fertilizer use in the apple orchard did not impact the yield physiological state of apple trees in the first year after application.

In 2022, the weather was more favorable (Supplementary Materials), and a higher yield was recorded in the T4, T3, and T2 variants, but these variants did not differ significantly from each other. Although the temperature was mild, there were periods of strong waterlogging at the end of July and in September. Apparently, the observed increase in the yield was due to the synergism of the relatively favorable weather, soil fertilizer application in the previous year, and foliar fertilization. The lowest yield under these conditions was in the control. Overall, micronutrient foliar fertilizion increased the yield by improving macronutrient uptake. For the three seasons of observation, the largest yield was noted in treatment T4; in the other treatments with foliar fertilization, the yield was lower, but this difference was not statistically significant.

## 4. Discussion

Weather conditions, particularly relative humidity, affect the permeability of the cuticle for metal ions (as the humidity increased from 50% to the saturation point, the permeability also increased) [57]. This could explain the different effects of the fertilizer spraying in the separate years of the study.

The foliar N observed in spring is likely shaped not by the conditions of the current season but mostly by the outcome of overwintering and the size of N reserves accumulated in the plant tissues during the previous season [58]. Later, foliar N begins to depend on the conditions of the current season which included experimental treatments such as foliar spraying.

Mn foliar fertilization displayed the greatest influence on the increase in foliar N in our study, both in the single-element treatment and in combination with other micronutrients.

Reports in the literature suggest that the application of Mn, Fe, and Zn boosts foliar N [59]. Vice versa, the application of urea stimulated the increase in foliar Mn [60]. Overall, the increase in foliar Mn was recorded in parallel with enhanced foliar N [61]. These facts corroborate the positive relationship between these elements found in our study. Neither Fe or Zn nor the other micronutrients in our study increased foliar N. In principle, Cu supplementation can be deteriorative for foliar N and P [62], but we did not observe such an effect.

We noted a trend of increase in foliar P content after Mn foliar fertilization in 2021 and 2022 (Figure 5) but not in 2020. This relationship also lacked a control.

According to the literature, the magnitude of micronutrient spraying on foliar P depends on the cultivar [41], but it is observed even on soils with low P availability [63,64], likely due to the mobilization of P by carboxylates excreted into the soil by the roots. However, too high a soil P content can interfere with the uptake of Mn from the soil [63]. There were also cases of other species in which the foliar application of Mn did not stimulate foliar P [65,66].

High constitutional foliar Mn (as in the season 2022) had no significant effect on foliar P (it was the same as in 2020 and 2021) and N (it was even lower than in 2020 and 2021). In view of this, the observed foliar N and P increase was indeed due to foliar Mn spraying. In 2022, we recorded the highest foliar Mn content. In this year the precipitation over the observation period was 515.9 mm, which was the highest throughout the study period (242.7 mm for the season 2020 and 325.5 mm for the season 2021). Earlier, Sud et al. [67] reported that Mn content in green tea shoots had a significant positive correlation with cumulative rainfall.

There are reports on the enhancement of foliar K by foliar Zn treatments in pistachio [68] and Granny Smith apples [69] but not in pomelos [65]. Spraying with Mn and Zn exerted the most significant (stimulatory) effect on foliar N, P, and K contents. Not all of the microelements affected the macronutrient uptake in our case; a plausible reason for this is that the spraying missed the period of highest demand for microelements by the apple trees. By the way, it is difficult to suggest such a period beforehand, so it will be the aim of our further research.

Finally, soil acidity is a key factor affecting the availability of nutrients contained therein. The optimum range of pH for N and K uptake is 6.0–6.5 [70], and for P uptake, it is 5.5–7.0 [71]. An increase in soil acidity because of chemical fertilization reduces the availability of key macronutrients. Soil micronutrients are also absorbed mostly at a pH above 7.0. Thus, microelement foliar fertilization helps to increase the bioavailability of macronutrients in slightly acidic soil.

## 5. Conclusions

We can summarize that single-element Mn spraying showed a larger effect on foliar N than mixtures with other fertilizers. Also, Mn spraying stimulated an increase in foliar P. The foliar K contents increased steadily in May–June after Zn spraying and also when Zn was added to the complete fertilizer mixture for spraying. The magnitude of the increase varied in the different seasons.

The possibility of increasing the main nutrient absorption of primary nutrients due to foliar micronutrient fertilization can be a basis for optimizing the nutrition of plants in orchards with a decrease in the rates of mineral fertilizers. This approach still needs to

be further scrutinized in terms of application timing and the plant demands of different micronutrients as a function of its growth stage varying during the vegetation period.

A key driver of the implementation of this approach is the maintenance of favorable soil pH which can be disturbed by the overapplication of primary nutrients and a decline in their availability for root uptake. Our results showed that the initial application of mineral fertilizers to the soil reduced its $pH_{KCl}$, which was subsequently restored. Applying chemical fertilizers at a lower rate (e.g., every 5 years) helps to maintain the $pH_{KCl}$ level optimal for root activity and nutrient uptake. improving the sustainability and environmental safety of industrial orchards. This combination of soil and foliar fertilization, aimed at improving soil fertility potential, should be further investigated.

**Supplementary Materials:** The following supporting information can be downloaded at: https://www.mdpi.com/article/10.3390/horticulturae9101144/s1. Figure S1: Contents of nitrogen in apple leaves in 2020; Figure S2: Contents of nitrogen in apple leaves in 2021; Figure S3: Contents of nitrogen in apple leaves in 2021; Figure S4: Contents of phosphorus in apple leaves in 2020; Figure S5: Contents of phosphorus in apple leaves in 2021; Figure S6: Contents of phosphorus in apple leaves in 2022; Figure S7: Contents of potassium in apple leaves in 2020; Figure S8: Contents of potassium in apple leaves in 2021; Figure S9: Contents of potassium in apple leaves in 2022; Table S1: Content of calcium and micronutrients in apple leaves in 2020, C treatment; Table S2: Content of calcium and micronutrients in apple leaves in 2020, T1 treatment; Table S3: Content of calcium and micronutrients in apple leaves in 2020, T2 treatment; Table S4: Content of calcium and micronutrients in apple leaves in 2020, T3 treatment; Table S5: Content of calcium and micronutrients in apple leaves in 2020, T4 treatment; Table S6: Content of calcium and micronutrients in apple leaves in 2021, C treatment; Table S7: Content of calcium and micronutrients in apple leaves in 2021, T1 treatment; Table S8: Content of calcium and micronutrients in apple leaves in 2021, T2 treatment; Table S9: Content of calcium and micronutrients in apple leaves in 2021, T3 treatment; Table S10: Content of calcium and micronutrients in apple leaves in 2021, T4 treatment; Table S11: Content of calcium and micronutrients in apple leaves in 2022, C treatment; Table S12: Content of calcium and micronutrients in apple leaves in 2022, T1 treatment; Table S13: Content of calcium and micronutrients in apple leaves in 2022, T2 treatment; Table S14: Content of calcium and micronutrients in apple leaves in 2022, T3 treatment; Table S15: Content of calcium and micronutrients in apple leaves in 2022, T4 treatment; Protection_system_description.docs; Weather_report.docs; Leaf_Micronutrients_2020.xlsx; Leaf_Micronutrients_2021; Leaf_Micronutrients_2022.

**Author Contributions:** Conceptualization, A.I.K., N.Y.K. and A.E.S.; methodology, A.I.K.; software, A.V.K.; validation, A.M.K., L.V.S. and V.N.K.; formal analysis, A.I.K.; investigation, A.I.K. and N.Y.K.; and L.V.S. resources, A.V.K., N.Y.K. and V.N.K.; data curation, A.E.S.; writing—original draft preparation, A.I.K. and A.E.S.; writing—review and editing, all authors; visualization, A.I.K. and A.E.S. All authors have read and agreed to the published version of the manuscript.

**Funding:** This research received no external funding.

**Data Availability Statement:** The data are available from the corresponding author upon reasonable request.

**Acknowledgments:** The invaluable support of the "Dubovoye" company in carrying out the field experiments and the provision of agrochemicals is gratefully acknowledged.

**Conflicts of Interest:** The authors declare no conflict of interest.

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
