# Peer review of "Foliar Mn and Zn Treatments Improve Apple Tree Nutrition and Help to Maintain Favorable Soil pH"

_horticulturae, doi:10.3390/horticulturae9101144_

Round 1

Reviewer 1 Report

I find your work very remarkable, but there have been huge mistakes and deficiencies in your writing.

Corrections that need to be made are presented in detail below.

1)      The writing order of the sections in the article should be as follows.

Introduction>Materials and Methods>Results>Discussion>Conclusions

But I guess this article was written in the following by mistake

Introduction>Results>Discussion>Materials and Methods>Conclusions

2)      An abbreviation such as "417 tr ha-1" or any abbreviation must be clearly written for once in the first place in the manuscript. (417 trees ha-1)

3)      Table 2 presents the available plant nutrients in the soil. How, where and by which method (ICP or or any other method) the soil analysis is done should be written briefly in the text.

4)      Line 307-309 could not understood. This lines should write more clearly.

“We sample leaves in the spraying day about 1 hour before foliar fertilizing. Usually, we sampled leaves before the spraying, so the analyzes result from the day of the first treatment show start leaf manganese contents” (in the manuscript)

5)      Some results of the research were associated with the weather condition, but no citaitons were shared to prove this. If the weather is not as claimed, the associations will be wrong.

6)      Some of the results were shared in the Discussion section. According to the template, the discussion section should be as follows.

“4. Discussion

Authors should discuss the results and how they can be interpreted from the per-spective of previous studies and of the working hypotheses. The findings and their impli-cations should be discussed in the broadest context possible. Future research directions may also be highlighted.”

7)      In addition, references 2, 5 and 17 cannot be reached. References 6 and 16 were prepared in 6 and 16 Russian languages. It is not known whether the documents on the website were prepared by following any academic process. International article, proceeding and book etc. references should be added instead of removing them from the article.

8)      In the references section,

a)       the DOI information of any reference is not shared. The following information is available in the template:

“Include the digital object identifier (DOI) for all references where available”

b)      “Title of the article” is  it is sentence order. that is, the first letter of the article name should be capitalized and the other letters should be lowercase. (Template)

Author 1, A.B.; Author 2, C.D. Title of the article. Abbreviated Journal Name Year, Volume, page range. (Template)

c)       Punctuation should not be used after “Abbreviated Journal Name” but most references used "dot"

9)       The following sentence should be added at the bottom of the article: